# The Demographic Implication for Promoting Sponge City Initiatives in the Chinese Megacities: A Case of Wuhan

**Shan Zheng** [1,2] ◉, **Yuting Tang** [3,*] ◉, **Faith Ka Shun Chan** [3,4,5] ◉, **Liyong Cao** [6] **and Ruixiang Song** [7]

1   State Key Laboratory of Water Resources and Hydropower Engineering Science, School of Water Resources and Hydropower Engineering, Wuhan University, Wuhan 430072, China; zhengs@whu.edu.cn
2   Hubei Provincial Key Laboratory of Water System Science for Sponge City Construction, School of Water Resources and Hydropower Engineering, Wuhan University, Wuhan 430072, China
3   School of Geographical Sciences and Natural Environmental Research Group, University of Nottingham Ningbo China, Ningbo 315100, China; faith.chan@nottingham.edu.cn
4   School of Geography, University of Leeds, Leeds LS2 9JT, UK
5   Water@Leeds Research Institute, University of Leeds, Leeds LS2 9JT, UK
6   Sponge City Company, Wuhan Steel Green City Technology Development Co., Ltd., Wuhan 430083, China; xiaocao2008net@163.com
7   Powerchina Beijing Engineering Co., Ltd., 1 Dingfuzhuang West St., Chaoyang District, Beijing 100024, China; songrx@bjy.powerchina.cn
*   Correspondence: yu-ting.tang@nottingham.edu.cn

**Abstract:** Urbanisation and ever-intensified rainstorms exacerbated urban waterlogging in some Chinese cities. In 2013, the Chinese government proposed a nationwide initiative, Sponge City, for managing the flood risk using the nature-based solution (NBS) approach. Pilot projects have been implemented among thirty selected cities, including Wuhan. Because the effectiveness of implementing NBS relies on the participation of the well-informed public, this study aims at identifying the factors affecting the awareness of the public about the Sponge City program. The viewpoint of people in Wuhan on urban floods and the Sponge City initiatives was surveyed among 1600 participants using a face-to-face questionnaire in mostly Wuchang area of Wuhan; more than 900 of them were further interviewed. The majority of participants, though recognising the threats from flooding, were lacking awareness and understanding of the Sponge City initiatives. The *Chi*-square analyses of association revealed that the level of awareness is affected by education, age and residential time; these demographic factors also affected their interpretation of the direct experiences of the water environment and governmental water management. To optimise communicating the relevant policy to the public, the content and the advertising tools for promoting Sponge City may need to be mindfully customised for targeted demographic groups.

**Keywords:** Sponge City; nature-based solution; urban flooding; public participation; environmental perception





## 1. Introduction

Environmental perception of the benefits and inconveniences brought by environmental changes may affect the level of support for specific environmental policies. People from a variety of sociocultural backgrounds, nevertheless, generate perceptions of the same changes that may potentially contrast with each other [1].

Nature-based solutions and low-impact development have been promoted and implemented in Western developed countries (i.e., European Union). This development has been considered progressive and was marked as a paradigm shift of urban water management when it comes to planning [2] in order to deliver multiple benefits, such as improving urban ecosystem services and improving green/blue-green spaces and social wellbeing (via more recreation facilities), other than urban flood control. Often, stakeholders and

residents are involved in such developments in the areas affected. This also fits the current wave of environmental governance that encourages public participation.

Recently, the urban water management issues in China have raised alarm by the Central National Government after years of flood events in big cities. The concept of building Sponge Cities (established in 2013) is as follows: Sponge Cities are equipped with the functions that absorb excessive water input to the areas while releasing the preserved water during drought conditions. The ideas of nature-based solutions and low-impact development have been widely implemented in the Sponge City infrastructures (e.g., via Sponge Parks, urban wetlands, forest, retention swales, etc.) [3]. The Sponge City program (SCP) has been implemented for more than 8 years; based on which, pilot projects in 30 selected pilot Sponge Cities in China have been successfully established. However, currently, it is a curious question, under the sociocultural context of China, as to how the demographic factors affect the perception of citizens in Chinese cities and their enthusiasm for participating in the Sponge City program [4].

Using Wuhan, a megacity in the middle part of the Yangtze River basin, as the case, this study aims to explore the influence of demographic characteristics in Wuhan on the understanding of the flood, the perception of the water management project, mainly the Sponge city project, and the level of the support. Some insights may be derived from the results of delivering the SCP effectively under the sociocultural context of Wuhan, so as to achieve better long-term sustainable urban planning in other Chinese cities alike.

## 2. Literature Background

### 2.1. Demographic Factors and Environmental Perception

Public support has been recognised as an important force to ensure that a well-intended environmental policy achieves its objectives [5]. Levels of environmental concerns and pro-environmental behaviours are reported to be affected by demographic factors in ways that may be related to the social norm and cultural context in regions and countries [6–11]. Understanding demographic influence on public environmental perception may guide policymakers in strategically encouraging public participation in order to achieve the objectives of promoting environmental policy, e.g., [12–14]. Studying the demographic effects on the specific population that the policy is targeting may offer more useful pieces of information than transplanting the results from case studies in other regions at various developmental statuses [7].

Rooted in speedy urbanisation and industrialisation, the inter-relationship between society and the physical environment has been altered in China, e.g., [15,16]; as a result, the environmental perception of the Chinese people has been shifting. Between 1998 and 2007, an unneglectable increased percentage (46% to 60%) of people would trade economic development to better protect the environment [17].

On the other hand, the reliance on governmental command is still strong and the sense that the public should be an important part of environmental governance remains weak, e.g., [18]. This trend is divergent from that among the European and North American countries during the prime time of their economic development.

### 2.2. Nature-Based Solution and a "Sponge City"

Flood is an environmental issue of global importance. Recent decades have seen a paradigm shift in flood risk management: nature-based solutions (NBS) have been increasingly embraced as part of the strategy. NBS can compensate for some shortcomings stemming from the engineered traditional grey infrastructure such as concreted embankments, dykes and dams [19–21]. Similar approaches are named low-impact developments (LIDs) in the US or sustainable urban drainage systems (SUDs) in the UK.

The nature-based design can reduce the impact of a wave during a storm surge at the seaside [19,21]; infrastructure such as bio-retention, green roofs, pervious pavements, bioswales and rain gardens can increase soil–water infiltration, recharge groundwater, protect the stream, purify the stormwater and enhance urban water quality for storage

and recreational purposes [22,23]. In addition to increasing the resilience of the urban environment to flood events by reducing urban runoff and peak flow, the infrastructure becomes part of the blue-green spaces in the inner-city that provide ecosystem service and improve the social wellbeing of the communities [24,25]. The communities can also benefit from the infrastructure during the period without floods.

Exacerbated by the climate-change-intensified rainstorms and by the urban-development-altered hydrological characteristics [26], the underequipped urban drainage systems in the old towns/districts (at the protection levels of the heavy rainstorm occurrence between 1-in-1 and 1-in-10 years) have made considerable socioeconomic impacts on the cities during a few flood events of catastrophic scale in China [27]. Some infamous cases include Wuhan 2016 flood, Guangzhou 2013 flood [28] and Beijing 2012 flood [29]. These floods represent the challenges that most of the Chinese megacities have been and will continue experiencing in flood risk management due to climate change [30].

Alerted by the situation, in the "13th Five years plan" (2016–2020), the National government of the Peoples' Republic of China declared the intention to address climate change and water issues (including water risk such as floods) targeting 1st and 2nd tier Chinese cities. The launch of the "Sponge City Program" (SCP) may be considered the positive response of the Chinese government to the recent global paradigm shift for flood risk management. The said effect of "sponge" was recommended to be created by making full use of green belts, the permeable pavement on roads and river systems, as well as allocating patches of blue-green space such as artificial wetlands and rain gardens [31] (MOHURD, 2014a); these spaces are designed to be assimilated into the local hydrological environment within the urban setting. This is in line with the idea of NBS. The political will to make this happen is reflected in the RMB 1.8 million investment from the Chinese government for designing and installing the NBS infrastructure in selected districts in more than 30 cities since 2015 [29,32].

The example of the Sponge City program development demonstrated that the technical approach of the nature-based solution has been gradually adopted by highly urbanised China to mitigate environmental issues. However, it is unclear how the environmental perception of the citizens affects the level of public participation under the Chinese socio-cultural context. Public support is rather important in successfully implementing the green infrastructure on which the Sponge City program heavily relies.

Quite a few studies have investigated public perception after 2015 and most of them were conducted in the US, UK or European countries [33]. Among these studies, the preferences of citizens and stakeholders have been emphasised, mostly based on the immediate physical experience (such as the greening areas or better air quality). Although these are co-benefits of the nature-based solutions for flood control, the preference does not seem to be based on its original purpose, which should have been flood mitigation. It is interesting how this decoupling of the citizen preference and flood mitigation performance affects the realisation of the intended results of flood mitigation. The answer to this is also very practical as the cost of building and maintaining the nature-based infrastructure is usually higher than just the greening infrastructures in the city. The expensive design for its flood control may be largely hidden underground and cannot be directly appreciated by the general public.

Further, experiences in developed countries showed that lacking communication with the public regarding the purpose and benefit of those infrastructures may reduce the future involvement or investment of the local communities in infrastructure maintenance [24], and the intended high cost-effectiveness may not be achieved [34]. However, with a slightly different mentality on who should be responsible for solving the environmental problems in China, it is unclear whether the environmental perception of the public may play a similar role in the effectiveness of nature-based infrastructures. Comprehending the attitudes, perceptions and willingness to participate of the public can be the key to designing, implementing and promoting the SCP.

## 3. Wuhan, Flood and Sponge City

The geographical location (located in Hubei Province, at the middle part of the Yangtze River Basin in China) of Wuhan makes it an important Chinese megacity and the national logistic centre for the national railway, road and river networks [35] (Figure 1A). It is now the capital of Hubei Province with registered residents of ~8.5 million. With a reputation as the River Town and the City of Hundreds of Lakes, its development is highly associated with the surrounding water bodies [36]. For example, Hanjiang River joins Yangtze River in the city centre of Wuhan, and the two rivers divide the central area into three parts of Wuchang, Hanyang and Hankou as seen from the traditional viewpoint of Wuhan citizens (Figure 1B,C). The area of Hankou is mainly the financial and business hub of the city, Hanyang is the industrial powerhouse and Wuchang is the education and research and development centre. The city development has followed this pattern though the revision of administrative boundaries of districts as the city expanded.

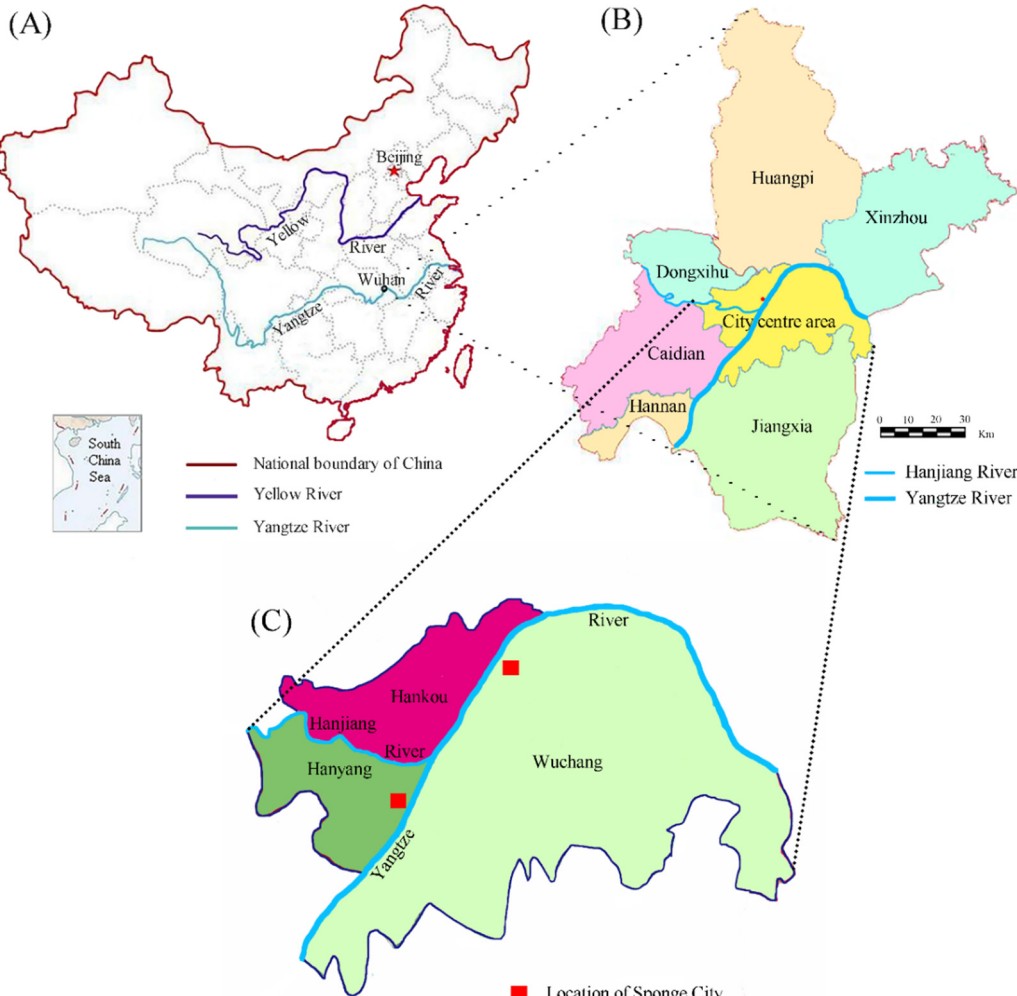

**Figure 1.** The study area; (**A**) the location of Wuhan City; (**B**) Wuhan about Yangtze River; (**C**) central districts in Wuhan about the pilot sites of Sponge City (source: Shan Zheng).

During the last few decades, several severe floods have potentially left a deep impression on the residents regarding the significant damages they have done to the city (e.g., the flood events in 1954, 1998 and 2016). Previous studies showed that the upward trend of the risk of surface water flooding due to the increased frequency of extreme weather [37] was exacerbated by continuous urban expansion. Wuhan Municipal Bureau of Statistics noted that rapid land use changes driven by urbanisation and socioeconomic development during the last two decades have reduced 34,000 ha of areas that were covered by surface

water (lakes and river channels) [38]; the capacity to accommodate the ever-increasing short-term intensive precipitation thus decreased.

The maximum daily precipitation at Wuhan city during 1951–2019 (data from http://data.cma.cn/, last accessed 28 February 2022) shows that the five greatest daily precipitation events occurred in 1959, 1982, 1998, 1969 and 2016, respectively. The maximum daily precipitation at Wuhan in 2011 and 2019 ranked the 9th and the 10th among those from 1951–2019, respectively. The relatively rich precipitation events in Wuhan in the recent decade have directed public attention to urban flood control at Wuhan (e.g., [39]). As indicated previously, although the precipitation and the formation of the flood are highly correlated, and it is easy for the public to connect the precipitation with a flood, factors such as land use [40,41], precipitation in the entire Yangtze River Basin and river channel management in wider regions [42] also contributed to whether the heavy precipitation accumulates in situ to form the flood. Further, the major flood events in Wuhan are usually part of the regional and cross-provincial flood events, not just the isolated event within the city. Thus, not all of these top ten daily precipitations generated noticeable floods in the records or news reports that we surveyed. The most reported floods in Wuhan during the past 50 years happened in 1998 and 2016, which coincide with the third highest and fifth highest maximum daily precipitation at Wuhan. The conditions and impacts of these two flood events are summarised (Table 1). These two major floods are relatively recent and the memories of the events may still be vivid in the citizens' memories.

**Table 1.** Flood status and impacts in Wuhan between 1998 and 2016 flood events [43,44].

| Flood Information | 1998 Flood | 2016 Flood | Comparison (Differences between 2016 and 1998 Floods) |
|---|---|---|---|
| Flood Type | Fluvial (Yangtze Catchment Flood) | Urban surface flood (Wuhan districts: e.g., Hongshan district, Nanhu district) | - |
| Total rainfall in Wuhan (mm) | 861.4 | 932.6 | +8.3% |
| Maximum weekly rainfall during the flood (mm) | 542.8 | 580 | +6.9% |
| Affected populations in Wuhan (×10,000 people) | 177.7 | 105.8 | −40.5% |
| Relocated populations (×10,000 people) | 16.3 | 20.7 | +27% |
| Flood enhanced property collapse (×10,000) | 1.6 | 1.01 | −36.9% |
| Flood enhanced property damages (×10,000) | 6.32 | 0.28 | −95.6% |
| Agricultural land damages (×10,000 hectares) | 233.42 | 230.9 | −1.1% |
| Direct economic impacts (×RMB 100 million) | 66.6 | 53 | −20.4% |
| Drainage power/capacity (×10,000 m$^3$/1 h) | 142.9 | 349.2 | +144.4% |
| Donations for flood relief (×RMB 10,000) | 19,000 | 4790 | −74.8 |

Wuhan became one of the 16 pilot cities in the first batch for implementing the Sponge City initiative in 2015 [45]. Since then, 455 infrastructures have been constructed, covering 38.5 square kilometres in locations mainly at two districts (Qingshan and Sixin) within the City Centre of Wuhan [36]. From an engineering and planning point of view, the flood mitigation capacity of these symbolic and progressive water management infrastructures installed within somewhat limited areas can only be discovered in the longer term. Despite the initial financial instalment from the central government, the long-term financial needs are still expected to be substantial as the design, construction and maintenance of the Sponge City infrastructures are expensive [46]. Cities under similar situations have been exploring the idea of acquiring financial support from private sectors or the public [47].

These geographical and social characteristics of Wuhan City have made the city a suitable site of investigation for the environmental perception of citizens on the environmental changes as well as their experiences and awareness about these newly installed nature-based infrastructures for SPC. Thus, this study aims to explore the level of understanding of the public on urban floods and Sponge City initiatives in Wuhan since the installation of the nature-based infrastructures, and whether the demographic characteristics of the city affected such understanding. The investigation may further contribute to the field of research about environmental perception under the context of Chinese culture. Insights may be derived for more effective delivery of the planning and implementation of green infrastructures such as the Sponge City infrastructures for similar cities. In the rest of this paper, we first explain the way this survey was conducted in Wuhan, the design of the survey questions, and the analytical methods used for the answers in the survey; the results are then presented together with our interpretation. We conclude by addressing how demographic compositions may affect the methods of raising environmental awareness, which leads to a recommendation on the ways to increase public awareness of the SCP in Wuhan to fulfil the objectives of sustaining the functional NBS.

## 4. Methods

A survey consisting of 22 single-answer multiple choice questions (single-answer MCQ, detailed in Section 4.1) and 9 interview questions (5 of which that are related to this study are listed in Section 4.2) was conducted in Wuhan city. The questions were designed to investigate the perception of local people on flood management and the newly implemented SCP.

The ethics procedure followed was approved by the University of Wuhan: the participants were informed of the interview; the concepts of Sponge City were explained to the interviewees when necessary, after which consent was obtained before the interview; the results were presented without revealing the identity of the respondents.

All interviews were conducted in public spaces where there is relative quiet to avoid as many interruptions as is reasonably possible (e.g., public parks and coffee shops). The survey was not conducted in private venues such as the accommodations of residents, as obtaining permission to enter private spaces may be complex and difficult and may raise health and safety concerns.

In total, 1688 questionnaires were successfully collected between July and September 2017 from the participants 18 years old and above. Only slightly over half of them offered to be interviewed for some or all of the nine interview questions (58% to 72% for each question, described in Section 4.2). The answers to those questions related to the aim of this study were analysed.

### 4.1. Questionnaire Design

The MCQ questionnaire is composed of five parts (from a. to e.):

a.  Demographic information (gender, age, education, years of living in Wuhan to represent the residential time; Q1 to Q4).
b.  Knowledge and perception about urban flooding and governmental water management (Q5 to Q7).
c.  Awareness and perception about Sponge City initiatives (Q8 to Q10).
d.  Perspective on climate change and local aquatic ecosystem (Q11 to Q14).
e.  Perceptions of and attitudes towards Sponge Cities and their environmental benefits and socioeconomic effects (Q15 to Q22)

In this study, the results of parts a., b. and c. (detailed contents of the questions can be found in the Appendix A) were subjected to statistical analyses (Section 4.3) to evaluate the understanding of water management and SCP of the general public.

*4.2. Structured Interview Surveys*

After the interviewee answered the questionnaires, if the participant agreed to continue, a set of nine questions was asked sequentially; answers in five of the questions were reviewed to aid in the understanding of the results of the questionnaire:

a.      What do you think about the urban flood (waterlogging) problem, do you think the Sponge City policy can relieve the flood risk of Wuhan?

b.      What do you think about climate change? Do you think the Sponge City policy can mitigate the impacts of climate change?

c.      Do you support Wuhan and other Chinese cities to carry on the SCP, for what reasons?

d.      Have you experienced the 1998 and 2016 Wuhan floods; would you share your experiences?

e.      Do you think the floods at Wuhan are enhanced by the force of nature or by a human? Anything to share about the differences in the past and present flood protection practices locally?

The interviews usually lasted between 10 and 15 min. The contents of the interviews were transcribed. The relevant quotes (originally in the Chinese language) and content translated from the replies of the interviewees were used to further analyse the possible reasons for the answer patterns we observed in the questionnaire.

*4.3. Questionnaire Survey Data Analyses*

4.3.1. Descriptive Analyses and Frequency Analyses

The answers to the questionnaires were subjected to frequency analyses: for every question, the number of responses under each answer option in a specific group of characters for our interests (e.g., age, education level and so on) was divided by the total number of valid responses in that subgroup to that question.

These ratios (frequencies) were then normalised in the form of percentages with a 95% confidence interval for comparisons between answer groups. The 95% confidence intervals of the percentages were estimated based on the equation: mean $\pm 1.96 \times [(x)(1-x)/n]^{(0.5)}$. Where x is the percentage and n is the sample size of the group of interests.

4.3.2. *Chi*-Square Analyses for Association

The *Chi*-square analyses for association were conducted to cross-examine the interactions between the demographic factors of the respondents (age, gender, education level and residential time in Wuhan) and the answers to the questions representing the general perception of a flood, and that of the SCP (categories were tabulated to be presented with the results).

The demographic factors were considered to be significantly associated with the ways respondents answered the questions when the results of the analyses showed a level of statistical significance of 0.05, i.e., (*p*-value < 0.05). Frequency analyses were conducted on the significant cases to identify the patterns of answers among subgroups.

The same analyses were conducted to further evaluate the connections between the answers in questions related to the perception of water management and those related to policy effectiveness (Q5, Q7, Q8 and Q9). This may aid in the interpretation of the underlying reasons for respondents to form their environmental perceptions.

**5. Results and Discussions**

*5.1. The City and the Participant*

The general geographic location of Wuhan has been reviewed in Section 3. For the sample collected in this survey, about 81% was carried out in the Wuchang area and another 12% in the areas of Hankou, Hanyang areas and the suburb combined (Figure 1C).

As a result, about 50% of questionnaires were distributed within the territories of universities, colleges, libraries and museums; 28% of the surveys were collected in populated indoor public spaces including railway stations, shopping centres, hospitals or restaurants, 19% in parks and 9% in residential neighbourhoods. It should be noted, the setting of the

Universities in Wuhan is more like a "college town"; within the territory of the universities, there are residential areas, business areas with shops and restaurants. People who pursue their daily life or live in these areas may not necessarily be the students or academic staff. Thus, 50% of the samples were taken from the territories of universities; this does not mean all 50% of the surveys were conducted on students, teachers or researchers. However, the percentage of people with higher education levels may be much higher in this area than in the general Wuhan metropolitan area.

As such, it is expected that the overall survey results reflect mostly the perception and opinions of the residents and visitors in Wuchang, part of the metropolitan Wuhan. Moreover, they do not necessarily need to have visited the Sponge City site infrastructure.

*5.2. Demographic Characteristics of the Respondents and Wuhan City*

Table 2 shows the demographic distribution of the interviewees in this survey. Among the respondents, the ratio of males over females (55.9% to 44.1%) is slightly higher than that of the population in the entire Wuhan City (51% to 49%, Table 2). The percentage (63%) of the respondents younger than 30 years old is much higher than that of the population in Wuhan (39.2% of people between 18 and 30 out of the city population of 18 and above). A disproportionally high rate (75%) of the respondents had received higher education. Combining these two factors and the locations where most of the interviews took place (Section 5.1), it is suggested that the overall statistical results may reflect the opinions in particular of Wuchang district, where the functions of the district are to support the operation of higher education as well as research and development.

**Table 2.** A comparison of demographic distribution for surveyed participants and Wuhan population.

| Independent Variable | Frequency | |
|:---:|:---:|:---:|
| | **This Survey (n)** | **Wuhan City in 2016 *** |
| Gender | | |
| Male | 55.9% (944) | 51.0% |
| Female | 44.1% (744) | 49.0% |
| **Age **** | | |
| 18–19 | 23.2% (392) | 1.7% |
| 20–29 | 39.8% (672) | 18.1% |
| 30–39 | 14.0% (237) | 19.4% |
| 40–49 | 13.6% (230) | 18.3% |
| >50 | 9.3% (157) | 42.5% |
| **Education level** | | |
| Primary school or below | 3.5% (59) | NA |
| Secondary and high school | 21.0% (352) | NA |
| Undergraduate | 66.0% (1109) | NA |
| Postgraduate | 9.2% (155) | NA |
| Unknown | 0.3% (8) | NA |
| **Residential Time (Years living in Wuhan)** | | |
| 0–2 | 24.4% (413) | NA |
| 2–5 | 23.7% (401) | NA |
| 5–10 | 11.3% (189) | NA |
| 10–20 | 18.1% (305) | NA |
| >20 | 22.2% (375) | NA |
| Unknown | 0.2% (5) | NA |

* The percentages were calculated based on Wuhan Statistical Bureau (2017), *Statistical Yearbook 2016* (http://tjj.wuhan.gov.cn/, last accessed 28 February 2022). ** Excluding the population age under 18.

As the sample sizes of the subgroups categorised based on demographic factors are sufficient (more than 50 questionnaires under each demographic subgroup, for most of the subgroup, the sample size exceeds 100), a cross-comparison between the perceptions of subgroups is feasible. Thus, we focus on analysing the potential influence of demo-

graphic factors on the environmental perception of the water environment and the newly implemented SCP.

Further, almost 50% of the respondents in this survey have lived in Wuhan for less than 5 years while more than 20% of the people have lived in the city for more than 20 years. The typical two-peak distribution of the residential time of the survey reflected that Wuhan is a long-established settlement with a significant number of permanent residents while in recent decades, immigrants and students pursuing higher education have flooded into the city for better opportunities and career development (see further analyses between age and residential time in the next section). Especially in the Wuchang District, where most of the surveys took place, the influx of young students may be obvious and significant.

*5.3. Influence of Socioeconomic Factors/Differences*

To cross-examine the correlation (or lack of correlation) between the perception of Wuhan people and the demographic factors, *Chi*-square tests of association were conducted (Table 3). Our ensuing interpretation of the causal association was established on two aspects: (1) quantitatively, we have a good size of samples under each subgroup (Table 2) to ensure reasonable confidence intervals of the frequency estimation; (2) logically, the demographic identity exists before the individual generates his or her opinions expressed in this interview and cannot be changed by the opinion expressed in the survey. Thus, when a significant correlation is observed, demographic factors are likely to be the explanatory (independent) variable for the survey answers.

**Table 3.** The effects of demographic factors on interviewees' perception.

| Categories | | Questions | Effect of Demographical Factors (Yes, *p*-Value of *Chi*-Square Test < 0.05; No, $\geq$0.05) | | | |
|---|---|---|---|---|---|---|
| | | | Gender | Age | Residential Time | Edu |
| General Perception of Flood (part b) | 5 | The seriousness of flood | No | Yes | Yes | Yes |
| | 6 | The cause of flood | No | No | No | Yes |
| | 7 | Effectiveness of governmental water management | No | No | No | Yes |
| General Perception of Sponge City (part c) | 8 | Do you know about Sponge City | Yes | Yes | Yes | Yes |
| | 8a | How do you know | Yes | Yes | Yes | Yes |
| | 9 | Does Sponge City Work | Yes | No | No | No |
| | 10 | Should this be duplicated in other cities | Yes | No | No | No |

Among the four demographic factors analysed, the education levels tend to affect the answers to most of the questions related to the perception of flood and the views on the Sponge City policy. Age and residential time affected the answering patterns in the same questions related to direct perception about flood and the associated policy (Q5 and Q8). The factors did not affect the understanding of the cause of the flood (Q6) or the effectiveness and knowledge of the governmental policies (Q7 and Q9). A cross-analysis showed that the respondents who reported older ages tended to have lived in Wuhan for a proportionally longer time (Table 4) (rank correlation coefficient = 0.52, $p < 0.05$). The correlation is significant but not to the extent of considering the two factors interchangeable.

Fifty percent of the respondents under 20 have lived in Wuhan for 2.5 years or shorter. This means more than half of them came to Wuhan after the SCP was officially implemented. However, also, 25% of the people who are younger than 20 have lived in Wuhan for the majority of their lives (15 years or above, Table 4). This group of young people would have experienced and observed the transition of the city. The majority (more than 75%) of the respondents aged above 50, by contrast, have had more than 17 years to experience the local environment; any changes brought by SCP may be new but are only a small part of their local memories.

**Table 4.** The relationship between age and residential time.

| Age (Category) | N | Residential Time (Year) | | | |
|:---:|:---:|:---:|:---:|:---:|:---:|
| | | 25th Percentile | 50th Percentile | 75th Percentile | Average |
| A (<20) | 386 | 1 | 2.5 | 15 | 7.01 |
| B (20 to <30) | 667 | 2 | 3 | 6 | 6.44 |
| C (30 to <40) | 235 | 7 | 12 | 22 | 15.46 |
| D (40 to <50) | 225 | 15 | 23 | 40 | 26.84 |
| E (50 and above) | 154 | 17 | 38.5 | 54 | 36.33 |
| Overall | 1678 | 3 | 6 | 20 | 13.39 |

Finally, gender seems to be less influential on the perception of flood than on that of the specific view of the policymaking (Table 2).

### 5.3.1. Cross-Analyses on Education Level

The frequency analysis results from each subgroup (Figure 2A–D) indicated that the relative proportions among answer options are not completely altered by the education levels; the absolute values of percentages between the same answer options in each subgroup, however, varied widely enough to cause statistical significance. For example, on the severity of the flood (Figure 2A), the proportions of the respondents under each subgroup believing that the flood is "*not serious*" are the lowest among the four answer options, but the absolute percentage is distinctively higher in the group with only primary education ($15 \pm 5\%$, other groups lower than 5% with confidence interval smaller than 3%); more than half of the respondents in each subgroup considered the flooding to be a moderate or serious issue.

With statistical significance, education levels are the only demographic factor that influences the way people think about the causes of the flood ($Chi = 21.84$, df = 9, $p < 0.001$). Three-fifths ($61 \pm 8\%$) of the respondents with the post-graduate level of education recognised that floods are caused by multiple factors, while only about half of the respondents in the rest of the groups think so ($52 \pm 2\%$, university, $49 \pm 5\%$ secondary, $53 \pm 13\%$ primary education) (Figure 2B). The connection between climate change and flood has received very little attention across the four groups with a variety of education levels.

It is worth noting that though more than half ($43 \pm 8\%$ plus $14 \pm 5\%$) of the respondents who received or are receiving post-graduate education considered the government to have done a moderate or significant amount of work, almost half of this group ($47 \pm 8\%$) is completely unaware of the implementation of the SCP; the lack of awareness of the program is even more obvious among other subgroups ($58 \pm 3\%$ among the university level of education, $58 \pm 3\%$ secondary and $63 \pm 12\%$ primary).

### 5.3.2. Cross-Analyses on Age and Residential Period

Age and residential time often affected the pattern of how the interviewees answered the same questions (Table 3). The respondents who lived in Wuhan longer or are older tend to consider flood a moderate and significant issue of the city (Figure 3A, $Chi = 49.46$, df = 12 and $p < 0.001$, and Figure 3B, $Chi = 45.01$, df = 12, $p < 0.001$). Although the SCP was not announced till 2014, allowing most of the respondents equal opportunities to be exposed to the news, the level of awareness of the program was still increased in the groups of longer residential time or the groups of elder ages (Figure 4A, $Chi = 45.20$, df = 12 and $p < 0.001$; Figure 4B, $Chi = 37.96$, df = 12 and $p < 0.001$). The levels of environmental awareness among the participants seemed to increase when life experience accumulated or attachment to the surroundings potentially grew. We later elaborate on the implication of this in the use of media for broadcasting the information related to Sponge City (Section 5.6).

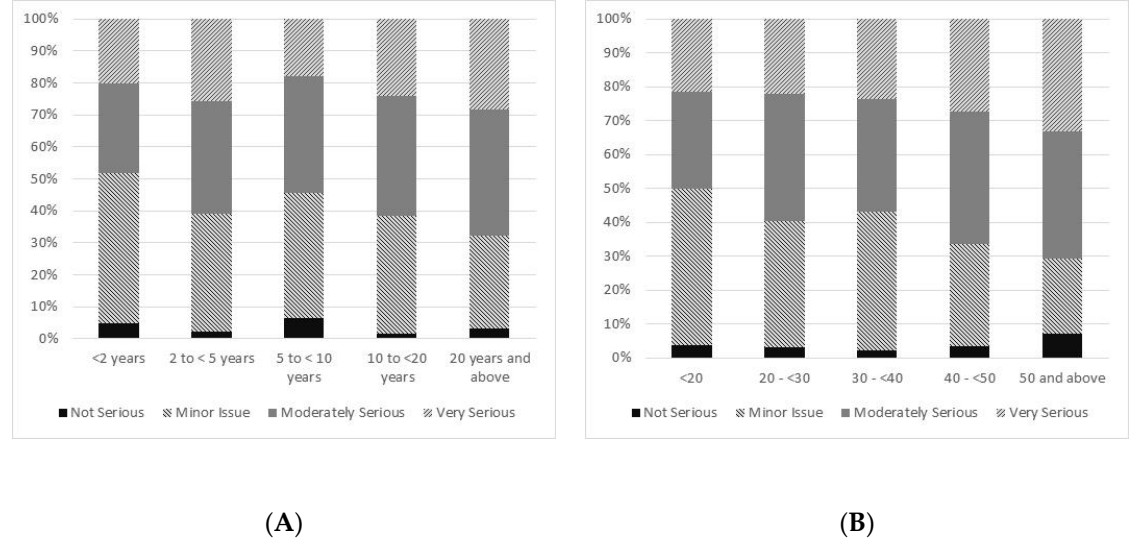

**Figure 2.** Cross-analyses between education level and survey questions: (**A**) the awareness of the severity of flood in Wuhan; (**B**) the perception of the causes of flood; (**C**) the perception of governmental efforts in flood control; (**D**) the awareness of Sponge City projects.

**Figure 3.** The effect of residential time (**A**) and age (**B**) on the perception of the severity of flood in Wuhan.

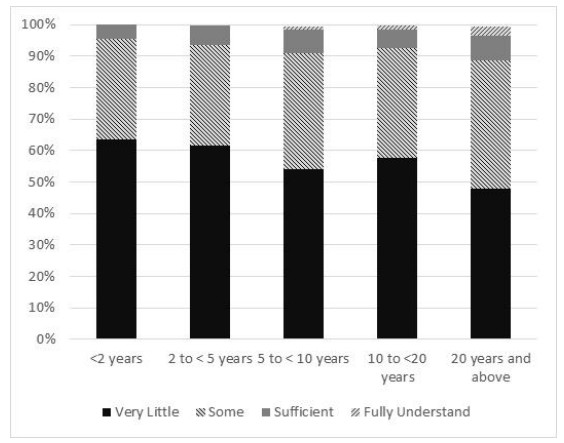 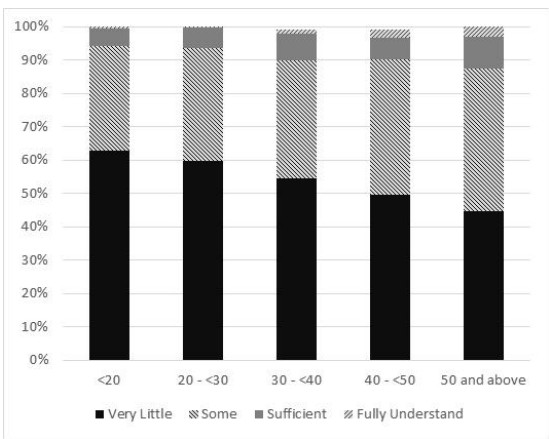

(**A**)                                                                 (**B**)

**Figure 4.** The effect of age (**A**) and residential time (**B**) on the awareness of Sponge City program.

The length of the residential period may affect one's view and knowledge of environmental transition in a specific place [48]. Whereas age may affect the view and the evaluation of the environment based on the variety and cumulative experiences of interactions between the person and the physical environments where he/she has been (e.g., part of the statistical analyses in [14] and [49]).

In our study, the elderly respondents are usually the residents who have lived in the area for a prolonged period (Table 4). Their observation of the environment may have been primarily based on the environmental condition in Wuhan. On the other hand, large numbers of the interviewees in the age group between 20 and 30 are likely to be university students, migrant workers or tourists who are relatively new to the city (more than half stayed less than 2.5 years). For the group younger than 20, the respondents may be the mix of students who recently came to Wuhan and the teenagers who have been in Wuhan since they were children, judging by their residential time (Section 5.3, Table 4): some are quite similar to, others are longer than, that of the group between 20 and 30 years old (25% of the group under 20 has lived in Wuhan for 15 years or longer while the same percentage of people in the group aged between 20 and 30 have only lived in Wuhan for 6 years or longer, Table 4).

The views of the two younger groups might be based on their comparison with the experiences from where they originally come, not necessarily Wuhan. Further, the age and residential time affected people's view on the seriousness of flood and the awareness of the government policy such as SCP but did not affect the knowledge of the cause of the flood and the evaluation of the government efforts. This implies that age and residential time influence perception that is more experiential than information based. Unlike education level, the two factors did not affect the way of answering the questions that require the understanding of the rationale behind a phenomenon (such as the cause of flood).

5.3.3. Cross-Analyses on Gender

Interestingly, gender affected all the questions related to the understanding of the Sponge City program (Table 3) and made no difference in the way people answered the questions related to flood conditions. Female respondents appear to be less confident about their levels of understanding of the program (Figure 5A, 9% more of the female respondents considered they know very little about Sponge City than the male respondents, $Chi$ = 19.07, df = 3, $p < 0.001$, Cramer's V = 0.11), but are slightly more generous when evaluating the effectiveness of the program (Figure 5B, 3% less of the female correspondents consider the SCP to have little effect, $Chi$ = 8.01, df = 3, $p < 0.05$, Cramer's V = 0.07). They entertained

the idea that the program can be implemented elsewhere more than the male (Figure 5C, *Chi* = 4.19, df = 1, $p < 0.05$, Cramer's V = 0.05).

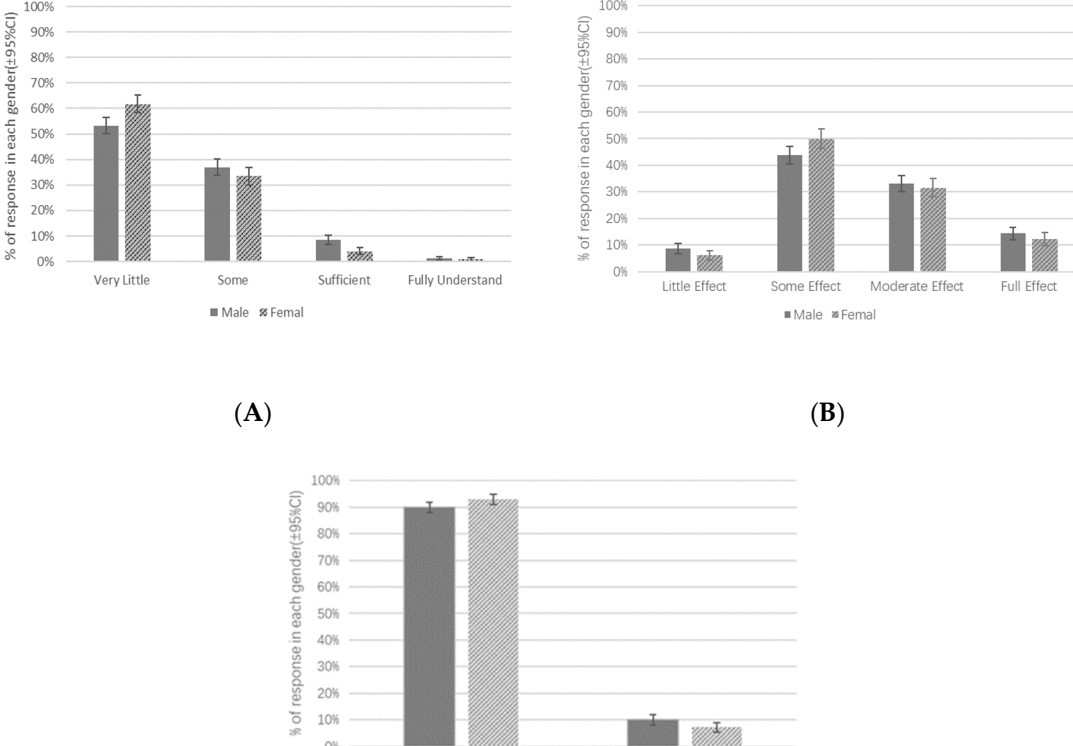

**(A)**                                        **(B)**

**(C)**

**Figure 5.** The effect of gender on the perception of the Sponge City program: (**A**) the awareness of the program; (**B**) the effectiveness of the program; (**C**) whether the program should be implemented in other cities.

Technically, the larger sample numbers within each gender group (Table 2) in the *Chi*-square tests narrow the confidence intervals of estimated percentages in the analyses. This increases the likelihood of detecting statistical significance between groups even when the differences are small. The diagnostic parameter Cramer's V (<0.1, [50]) indicated that the effect sizes of significant results are small for most of the analyses related to gender in our study. Thus, it is considered that, although gender is an influential factor in the environmental perception of the water environment in Wuhan, the effect of gender may not be very big.

*5.4. Qualitative Evaluation Based on Interview*

5.4.1. The Response Rate about Demographic Background

We attempted to collect detailed viewpoints on the flood and SCP that may not have been captured in the questionnaire by conducting interviews. Based on the statistics of who responded to the interview questions, we evaluated whether any specific demographic groups are more willing to express their opinions about the flood and SCP.

Overall, for each question, we had about 70% of people (between 72% and 69%) who were willing to be interviewed after finishing the questionnaire questions. The rate of answering slightly decreased as the question number increased, implying the interviewees might have felt that the survey took too long and then they gradually lost interest.

Most of the responding rates for subgroups under demographic categories are between 65% and 75%. However, the responding rate for the subgroups with primary or lower

education levels is lower than 65%, implying this group may be slightly more hesitant in elaborating their view. Based on this statistic, it is considered that in general, the demographic factors may not vary the level of motivation regarding people expressing their ideas about the flood unless their education levels are much lower than average. The detailed text analyses of each interview script are beyond the scope of this paper, but to verify the consistency between the content of the interview and the results of the questionnaire, general reading of the interview scripts is summarised (in the two following sections).

### 5.4.2. Experiential Understanding of the Causes of Flood

The interview records show that many interviewees (650 out of 990) perceived multiple factors that caused an urban flood in Wuhan (over 50% in almost all the subgroups), consistent with the records on structured questionnaires (Section 5.3).

The content in the interview further illustrated that this understanding of the interviewees is mostly empirical. They experienced the intensive rainstorms, felt the impacts of climate change and observed dysfunction of the land drainage protection system; more than 30% of respondents (340 out of 995) implied the current land drainage system is not robust enough to protect Wuhan from the floods; some respondents reflected on the topographical character of the city that exacerbates the situation: " . . . *the town centre of Wuhan is located on the low-lying area where waterlogging during the rainy season is easy to occur . . .* "

With the increased areas of artificial impermeable pavement, the city grew dependent on manmade drainage for discharging stormwater. Unfortunately, the drainage system in most Chinese cities, including Wuhan, is only equipped with the flood defence infrastructure at the level of the 1-in-1 year or up to 1-in-10 years return period for rainstorms [29,51], not enough for facing the increasing frequencies of intensive precipitation. Wuhan was flooded quite easily even at the scale of rainstorms smaller than the one in 2016, during which the city was inundated by the recorded high precipitation of 932.6 mm in total (increased 8.3% compared to 1998 flood). This phenomenon was observed by the participants:

> *"Wuhan has developed very fast these days and relying on drainage to relieve urban floods, but the drainage system is not able to deal with the big storms such as the 2016 flood, plus other events . . . ";* " . . . *Wuhan has been urbanised quite a lot; the green spaces for catching rainwater is decreasing; plus the lakes, wetlands and ponds are diminishing. I think these facts cause the urban floods in the city . . . "*

The perception of what triggers the flood based on the general experiences expressed by interviewees who took the interview reflects the reality in Wuhan adequately. However, individual interviewees may not point out all the potential factors; their observation may be accurate but partial and incomplete.

### 5.4.3. The Awareness of Sponge City Program

The questionnaire survey indicated that within each subgroup by the demographic categories, between 45% and 63% of respondents do not consider that they know the Sponge City program very well; this is consistent with the finding in the interviews. Quite often, the interviewees replied to the related questions with some words like: "*No, I don't know too much about the program . . .* ", or "sorry I have not heard too much about it . . . " Admitting the lack of knowledge, a participant still said " . . . *maybe it (the program) is useful for relieving flood issues . . .* " Indeed, most of the participants believed that the program can mitigate the flood risk for Wuhan. For the interviews, 720 people out of 995 people declared their support. A small portion (30 out of 995) supported the program clearly out of trust in the authority of the government. This positive attitude towards the program may be transformed into high levels of compliance with the related policies enforced.

With their empirical understanding of floods (180 people interviewed had experienced the major event in 1998), some interviewees hoped that the program can alleviate the risks resulting from the combination of strong rainstorms and weak drainage systems.

> *"[SCP] absolutely can relief the urban flood issues in Wuhan . . . "; " . . . I do not understand what is the exact meaning of 'Sponge City Program', but I hope for any new policy that solves the urban flood problem in Wuhan . . . "*

Others expressed doubts:

> *"I think the SCP cannot solve the issue like a flood in July 2016. Eventually, we need a better land drainage system . . . "; "the program sounds like a nice idea, but . . . it only covered a rather small area in Wuhan. Promoting the program to be implemented everywhere should be very difficult . . . "*

The design of the SCP aims to address the rainfall that is within 1-in-30 years for 24 h [52]; this is already an improvement to the current capacity for accommodating the precipitation, but the design is more useful to address the frequent smaller scale of flood, not the large-scale ones that result from intensive rainstorms (e.g., 2016 July storm at Wuhan; 2012 storm at Beijing; 2015 June storm in Shenzhen and Guangzhou) [29]. The interview conversations revealed mixed perceptions and expectations of what the program is capable of; the interviewees either retained wishful thinking that the program can completely solve the problem of a flood, or felt uncertain about what the associated NBS infrastructures can really do provided the limited coverage. Indeed, in a recent visit to a Sponge City pilot site in a residential area in Qingshan district (24 August 2019), the director of the project has explained to us that to make the small size infrastructure function more efficiently, the strategy of selecting the site for implementing SPC involves targeting the area with a high risk of flooding during the small or moderate precipitation event. These types of strategic planning of the infrastructure, accompanied by the clarification of the objectives, should be communicated to the general public to better manage expectations.

Most of the respondents considered that the program can bring added benefits such as improved living quality and river ecology:

> *"I support the Program, it seems to be very environmentally friendly"; "I think the SCP can contribute to green space development, benefiting the urban city like Wuhan and improving urban ecology."*

The program, viewed by some, may help control the increasingly warmer weather "by expanding green spaces". Others recognised that "climate change is a global issue", beyond the scope of the SCP.

Although 928 out of 982 people in the interview considered that the areas where the program is implemented should be expanded, the cost is also in the mind of some Wuhan people, " . . . *the Program is not cheap, and funds are expected from taxpayers; I hope it can perform well . . . "* Meanwhile, some are mindful regarding the time required to allow the positive outcomes to show: " . . . *I understand it is costly; it is a good concept and should be given enough time to let the effects show".*

Indeed, RMB 1.8 billion provided by the National Ministry of Housing for the pilot studies seems costly, but after being divided among 30 cities for spending during 2015–2018 [29], the amount can only allow most of the municipalities, Wuhan included, to test five to six sites [31,47]. The central government has set up a target that the infrastructures built under the program should be able to collect 70% of rainwater for a further 20% of urban areas by 2020; the proportion is to be increased to 80% by 2030 [52]. The scales of the current pilot sites are far from capable of achieving the objective. To extend the coverage of the Sponge infrastructure, especially for the areas where retrofitting the on-ground and underground infrastructures (i.e., properties, roads, cables, underground, water and energy pipes and sewers) is needed, financial support would be indispensable. The sources would be either from the municipal government, the public or stakeholders; the stakeholders are likely to be the private sectors that would benefit from the increased property values (price, ecosystem service or quality of life for the people) nearby such environmental improvement.

### 5.5. The Cross-Analyses between Questions

The cross-analyses between question 5 and question 7 (*Chi* = 43.9, df = 9, *p* < 0.001) showed that people who felt the flood is "not serious" and "extremely serious" are the two groups with higher percentages (12 ± 8% and 16 ± 4%, respectively) that considered that the government has done little to mitigate the flood risk (Figure 6A).

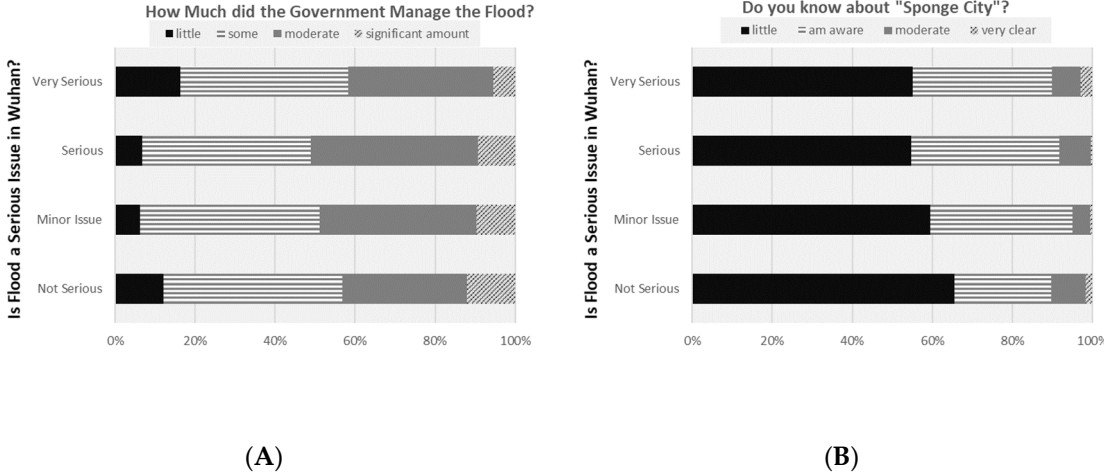

**(A)**                                              **(B)**

**Figure 6.** The relationship between the perception of the seriousness of flood (Q5) and (**A**) the perception of flood management conducted by the government (Q7); (**B**) the knowledge of Sponge City (Q8).

The underlying reasons for this shared view, however, could be quite dissimilar: people who do not consider flood an issue may not actively seek the related information. Their perception of governmental management might be changed easily after the information is shown to them. By contrast, people who consider flood a serious problem may have paid attention to what governments did, based on which, they concluded that the level of water management is insufficient. This presumption was verified by the fact that the percentage of the people who consider that government conducted a significant amount of flood management is the lowest (5 ± 2%) among the group who consider flood a serious issue. The percentage increased to the highest (12 ± 8%) in the group who consider flood in Wuhan not serious.

The percentage of respondents who claimed never to have heard of the term "Sponge City" gradually decreased as their evaluation of flood risk increased (Figure 6B, *Chi* = 30.38, df = 9, *p* < 0.001); this further verifies the idea that people considering flood a threat tend to pay attention to the related governmental projects but may not necessarily consider that the government has done enough.

Still, at least half of the respondents in the group who consider flood a serious problem have little knowledge about "Sponge City", implying a lot of work is to be done in promoting the program.

A relatively higher percentage (17 ± 17%) of people in the small group (n = 18) who considered themselves knowledgeable about "Sponge City" think the government did little in managing the flood, while this is also the group with the highest percentage (22 ± 19%) to say the government has conducted a significant amount of flood management (cross-analyses between Q7 and Q8, *Chi* = 61.17, df = 9, *p* < 0.001, Figure 7A). People who do not know "Sponge City" tend to consider the government only did very little (11 ± 2%) or only some work (48 ± 3%).

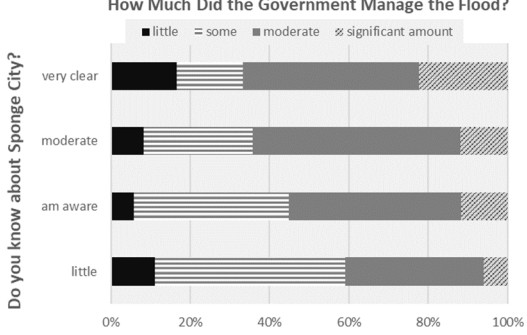 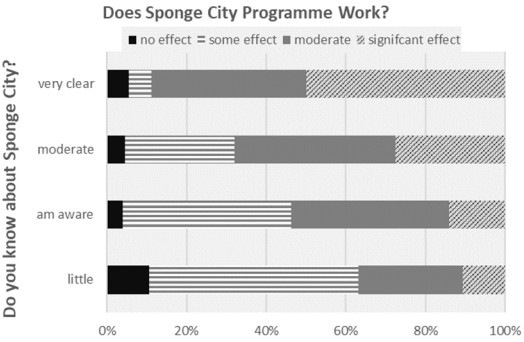

**(A)** **(B)**

**Figure 7.** The knowledge of Sponge City (Q8) about (**A**) the perception of the degree of flood management conducted by the government (Q7); (**B**) the perception of whether the program works (Q9).

Not surprisingly, the degree of the self-claimed knowledge about "Sponge City" is significantly related to their beliefs on the effect of "Sponge City" (cross-analyses between Q8 and question Q9, *Chi*-square=106.85, df = 9, $p < 0.001$). More than 50% of people in the group who say they fully understand "Sponge City" considered this project will be very effective. In the group who does not know about Sponge City, only less than 40% of the respondents (very effective $11 \pm 2\%$ and effective $26 \pm 3\%$) think it can be useful (Figure 7B). Thus, promoting SCP may include a series of advertising and education on the objectives and the associated concept of the design so that the public may appreciate the efforts the government has put in and the benefits of the program they might have enjoyed and will receive.

*5.6. The Media Public Received Information*

The demographic factors affected the ways respondents received the information related to SCP (Question 8a). Although TV is the major media from which respondents received the information, the percentages using TV were low among younger ages of respondents ($27 \pm 7\%$ for the group younger than 20; $30 \pm 5\%$ for the group between 20 and 30, Figure 8A), similar to the trend of the percentage of people who access the information by reading newspapers.

The decreased percentages among younger groups in using TV and newspapers were made up by the proportion who reported they obtained the information from the lectures ($37 \pm 7\%$ for the group younger than 20; $31 \pm 5\%$ for the group between 20 and 30, Figure 8A). Further, the respondents who reported staying in Wuhan for less than five years tended to read newspapers less but obtained the information from lectures (Figure 8B). This reflected the idea that respondents from these age groups are still college students, for whom the lectures may be a convenient way to learn about government policies. This conclusion is supported by the higher percentages of university-educated respondents who indicated that they learned about the SCP from the lectures (Figure 8C). In the cross-analyses between gender and the use of media, females tended to obtain the information from lectures while males received it from newspapers. Between 10% and 20% of respondents in subgroups of all demographic factors reported they obtain information from "other" types of media, implying the new ways of accessing the information may have gained popularity. These could be apps with news reporting functions on a mobile device. The details of these "other" categories warrant further investigation, both regarding what types of other media the users would use to obtain knowledge of Sponge Cities and regarding whether the information related to the Sponge Cities has been delivered via these new media. The results may inform the policy and decision makers of a more effective channel to promote the concept and the awareness of water management.

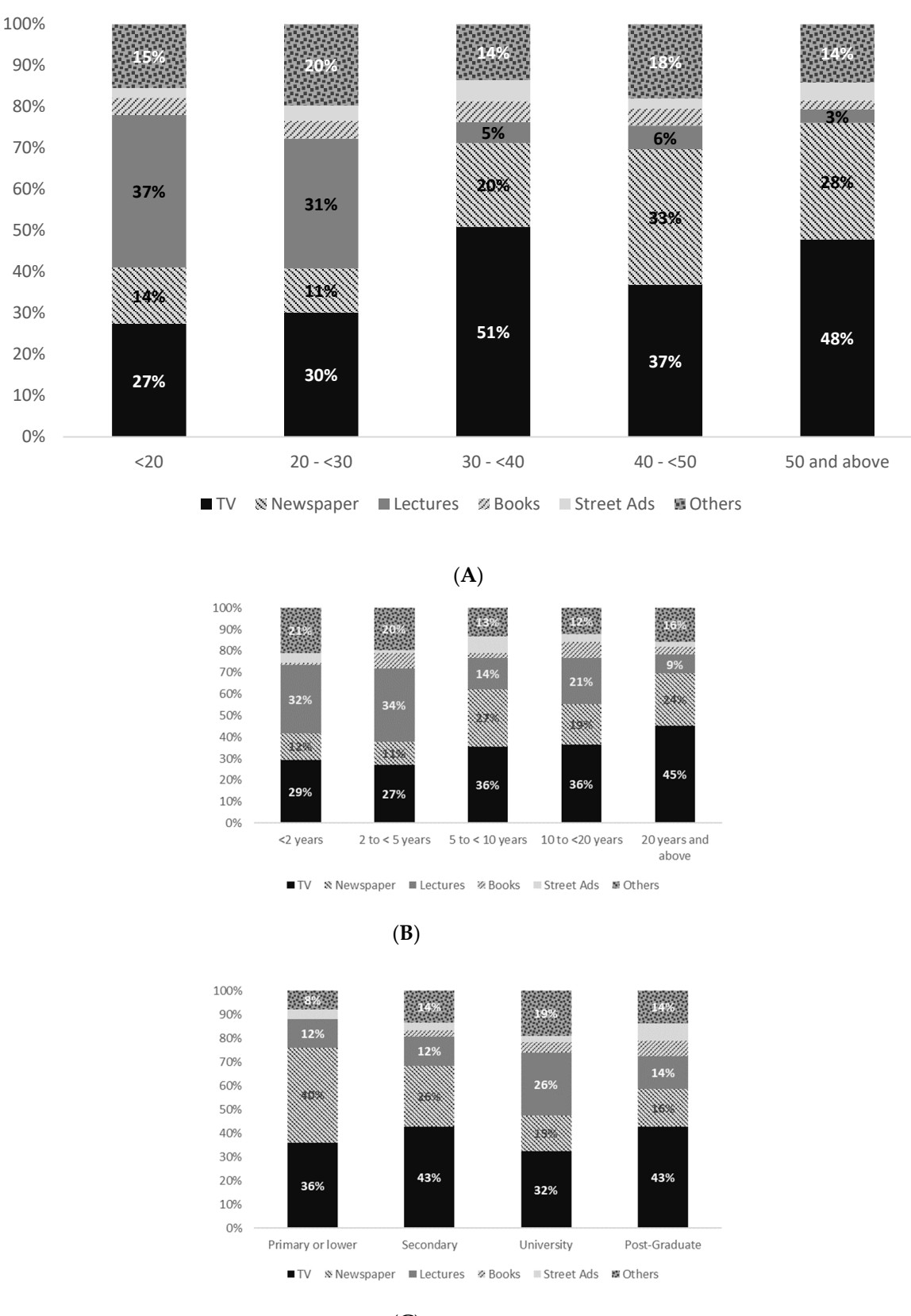

**Figure 8.** The means to access the knowledge related to Sponge City (Q8a) among (**A**) age groups; (**B**) residential time; (**C**) education level.

Additionally, our results regarding age and residential time indicated that younger people or people who stayed in Wuhan metropolitan areas for a shorter period tend to be less aware of the SCP. Thus, improving the awareness of SPC and flood management through school education may be an effective approach.

## 6. Discussion to Recommend Policy Dissemination and Further Research

Our survey was mostly conducted in Wuhan metropolitan areas, especially in the districts dominated by higher education institutes and residential areas. The relatively large sample size enabled us to evaluate the environmental perception of the subpopulation in Wuhan metropolitan areas based on a few demographic factors. Similar to the conclusions of some previous studies [49,53,54], the education level and local experiences are influential on the environmental perceptions. Respondents with higher education levels, who lived for a longer time in Wuhan or who have older ages, tend to recognise the seriousness of the flood that occurred in Wuhan; they also appeared to be more aware of the SCP. The ages and residential time, different from the education level, have not affected the way people consider the causes of the flood. Gender only affected the pattern of the answers in the questions related to Sponge City, not the experience with a flood. Based on these observations, recommendations related to policy dissemination and further research are made.

### 6.1. Experiential Perception and Educated Knowledge on SCP

The study revealed that in 2017, more than half of the individuals in each of the subgroups we analysed considered flood risk to be high in Wuhan; considering a serious flood had just happened in 2016 (Section 3), this result is not surprising. Whereas less than half of them are not aware of the SCP that was initiated in 2014. The understanding and participation of the public in the SCP is critical for the success of the program. Our cross-analyses suggest that people who experienced the local environment of Wuhan longer would have a deeper empirical understanding of the flood risk and would tend to be relatively aware of the SCP, the program designed to tackle the flood problems. The interview reveals that their experiential understanding usually reflects part of the real situation but may not be comprehensive. We consider that for people who develop their perception of flood and SCP via experience, the accessibility of the Sponge City infrastructure may be the most effective way for them to learn and comprehend the SCP. This idea may be verified by conducting further studies targeting the sites near the Sponge City infrastructure.

On the other hand, people with higher education levels may have understood the flood risk and the newly implemented SCP via their education institutes. The knowledge acquired in this way tends to cover multiple aspects; thus, people with higher education levels tend to recognise that a flood event usually results from the interaction of multiple factors. It is deduced that for people who develop their perception of flood and SCP via learning in school (not experiential), the accessibility of the related message may be the key to raising awareness and deepening understanding of the program. This may be similar to people who learn about the flood and SCP from news media.

### 6.2. Recommendation for Informing the Public

Our study has shown that demographic difference affected the understanding and level of appreciation of the water management program such as Sponge SCP. As the demographic backgrounds also affect how the participants access the related information (Section 5.6), in order to pursue environmental education and promote the SCP, governments need to select media with the backgrounds of target audiences in mind [55,56]. In the case of the Wuchang area in Wuhan, for example, it may be effective to spread the ideas of Sponge City in the higher education institutes to the students.

On the other hand, TV and newspapers are the major channels utilised by the residents over age 50 to obtain information. Considering the much larger percentage of people older

than 50 years old in the entire Wuhan City (Table 2, 42.5% in 2016) than the percentage of the same age group in this survey (9.3%), concentrating on Wuchang where higher education institutes are clustered, the newspaper and TV advertisements may be a more effective means of promoting flood management and SCP than our study showed for the wider population in Wuhan, as this age group is composed of more than 40% of the residents in Wuhan.

Further, the emerging news media and apps in the mobile device may have become non-ignorable tools used by the public to assess the information; the temporal variation of the behaviours in accessing the information may become another interesting and practical subject to investigate. We expect that this type of message delivery will become increasingly important for Chinese government to disseminate the public policy. It is suggested that delivering the information related to flood and SCP using popular mobile apps such as WeChat in China may become more and more effective.

Making the name and ideas of the SCP known to the public is only the first step for effective communication and education. It is important to manage the expectation of the outcomes after the SCP is implemented. In particular, the SCP is not a "Messiah" for flood control during intensive rainstorms such as the one exceeding precipitation of 100 mm/h. Our survey showed that in Wuhan, it seems people, in general, are aware of the flood risk and some may have had a higher expectation of what SCP may achieve in terms of managing the flood (Section 5.4.3); the campaign for the SCP may need to be performed in a more sophisticated way so that the expectation of benefits brought by this policy may become realistic. For example, although the precipitation in Wuhan was much heavier in the 2016 flood event that that of 2019, we saw lower percentages of population affected and a smaller scale of infrastructure damages as well as economic losses (Table 1). This somewhat demonstrates that water management in the area is advancing though the flood still occurred. It is the matter of how the management strategies, combining Sponge City and local grey infrastructure, can catch up with the increasing frequency and scale of climate extremes in order to prevent floods.

*6.3. Further Research and Limitations*

In this study, we identified the site-specific effects of the demographic factors on environmental perceptions of floods and the water management policy in Wuhan, China. On the subject of the influences of the demographic factors on environmental perceptions, among the population surveyed, we found the common demographic factors may not necessarily act in a way similar to previous studies outside of China and to some extent within China. Ding et al. studied and reviewed the willingness to pay for the SPC in a few Chinese cities; the effects of demographic factors such as age and education level are variable [57]. This is consistent with a previous study which indicated that the effects of demographic factors on pro-environmental behaviours or attitude may vary even at the level of towns within the same region: the tendency of growing attachment to a place, the length of residence and caring for the place, as well as the interaction between these variables, are changeable [48,58]. This reflects the site-specific nature of the policy influences which need to be investigated before making, promoting and implementing the environmental and water management policies to achieve the intended policy goals while avoiding the unintended consequences.

As indicated previously (Table 2), the population surveyed in the study may be skewed towards younger and educated groups of the cities. Our conclusion on the attitude of age groups and education levels relies on sufficient numbers of participants under each demographic group; this way, the confidence interval of our statistical analyses is within a reasonable level for us to identify the effects of the demographic backgrounds on the perception of the SCP. We recognise the limitation of extrapolating the conclusion to the entire Wuhan area, which covered some relatively rural areas with larger proportions of elderly people and with an average education level that may not be as high as the metropolitan area. On the other hand, currently, most of the SCP infrastructures in Wuhan

are located in metropolitan areas. Thus, the conclusion and recommendation of this study may still be meaningful and applicable.

## 7. Conclusions

In conclusion, the demographic factors such as age, gender, education level and residential period affected the environmental perception of people in Wuhan on flood-related subjects and the newly implemented SCP. Thus, for educating citizens on water management and promoting the SCP, the government may need to select effective means targeting the demographic groups either with larger population sizes or with lower levels of experiential understanding or educated knowledge about water management issues.

**Author Contributions:** S.Z., questionnaire design and organisation and original manuscript writing; Y.T., questionnaire design, statistical analyses and manuscript writing review and editing; F.K.S.C., interview data analyses and interpretation; L.C., interview guidance and organisation; R.S., review and editing manuscripts. All authors have read and agreed to the published version of the manuscript.

**Funding:** This study was also partly funded by the National Natural Science Foundation of China (Grant No. 52079095) and funded by Ningbo Science and Technology Bureau, Project titled "Quantifying the impact and risk of Typhoon related flooding in Ningbo (Grant No. 201401C5008005).

**Institutional Review Board Statement:** Not applicable.

**Informed Consent Statement:** This is not a medical study and informed consent was obtained from all subjects involved in the study. No information that makes identify any individual is disclosed in this work to maintain the confidentiality.

**Data Availability Statement:** The content of the questionnaire questions analysed in this manuscript has been inserted in Appendix A; a summary of questionnaire analysed in this manuscript is presented in Appendix B.

**Acknowledgments:** This research obtained support from Wuhan University and students from the Flood Control module for all the surveys conducted from the period of July to September 2017. Additionally, Ziyi Huang from Wuhan University and Xue Zhenyang from the University of Nottingham Ningbo China have contributed to the data organisation. Zhou Yunjin from Wuhan University is thanked for editing Figure 1.

**Conflicts of Interest:** All authors of this manuscript declare there is no conflict of interest in this study and the manuscript preparation. The funders had no role in the design of the study; in the collection, analyses, or interpretation of data; in the writing of the manuscript, or in the decision to publish the results.

## Appendix A

Questionnaires and Interview Related to this Study on Urban Waterlogging, Sponge City and River Ecological Restoration in Wuhan.

| Questionnaire ID: | Location: | Time: |
|---|---|---|

**Questionnaire on Urban Waterlogging, Sponge City and River Ecological Restoration in Wuhan**

(1) Age:
    a. Under 20 b. 20–29 c. 30–39 d. 40–49 e. Above 50
(2) Educational Background:
    a. Primary school or lower b. Secondary/High school c. Undergraduate school d. Post-graduate school
(3) Gender
    a. Male b. Female
(4) How long have you been in Wuhan?
    _______year(s)

(5) Do you think waterlogging in Wuhan is serious?
a. Not serious b. Minor issue c. Moderately serious d. Very serious
(6) What do you think causes waterlogging in Wuhan?
a. Heavy rains b. Global climate change c. Poorly designed urban drainage facilities d. Multiple factors
(7) Do you think the government has done enough to mitigate the urban waterlogging in Wuhan?
a. Little b. Some c. Moderate d. Significant
(8) Do you know the Sponge City Program (SCP) in Wuhan?
a. Little b. Aware c. Moderate d. very Clear
If you didn't choose **a**, through **which medium** did you learn about this project?
a. TV b. Newspapers c. Lectures taught by teachers d. Books e. Billboards f. Others
(9) Do you think SCP is effective in improving urban flood control capacity?
a. Not at all b. A little c. Medium d. Very much
(10) Wuhan is one of the first 16 pilot cities for SCP in China. Should this Sponge City construction be promoted in other cities in China?
a. Yes b. No

Additional questions in questionnaires (Q11 to Q 22) will be presented in the article specifically related to the contents and the analyses of the answers.

**Interviews on urban waterlogging and water ecology environment in Wuhan**

(1) What do you think about the urban flood (waterlogging) problem, do you think Sponge City policy can relieve the flood risk in Wuhan?
(2) What do you think about climate change? Do you think the Sponge City policy can mitigate the impacts of climate change?
(3) Do you support Wuhan and other cities to carry on the SCP, for what reasons?
(4) Have you experienced the 1998 and 2016 Wuhan floods; would you share your experience?
(5) Do you think the floods at Wuhan are enhanced by the force of nature or by a human? Anything to share about the differences in the past and present flood protection practices locally?

Additional Interview Questions (Q1, Q7 and Q9) will be presented in the article specifically related to the content and the analyses of the replies from the questions.

**Appendix B**

Summary of Questionnaire Results Related to this Study

**Table A1.** General information regarding data collection and participants.

| Sample Size n | Duration of Questionnaire | Categories * | AGE | EDUCATION | GENDER | Residential Time (Years) | |
|---|---|---|---|---|---|---|---|
| 1688 | 2 July to 25 August 2017 | A | 392 | 59 | 944 | Minimum | 0.0 |
| | | B | 672 | 352 | 744 | 25 Percentile | 3.0 |
| | | C | 237 | 351 | - | 50 Percentile | 6.0 |
| | | D | 230 | 155 | - | 75 Percentile | 20.0 |
| | | E | 157 | - | - | Maximum | 82.0 |

* The answer options under each category can be found in Appendix A—no option in this category is in the question.

**Table A2.** Summary of the frequency of answers for questions related to perceptions on flooding and Sponge Cities.

| Categories * | Flood Related | | | | Sponge City Related | | |
|:---:|:---:|:---:|:---:|:---:|:---:|:---:|:---:|
| | Q5 | Q6 | Q7 | Q8 | Q8a | Q9 | Q10 |
| A | 58 | 241 | 152 | 959 | 285 | 121 | 1465 |
| B | 633 | 61 | 728 | 596 | 145 | 737 | 142 |
| C | 592 | 509 | 658 | 109 | 172 | 514 | 0 |
| D | 405 | 877 | 145 | 18 | 33 | 214 | 0 |
| E | - | - | - | - | 28 | - | - |

* The answer options under each category can be found in Appendix A—no option in this category is in the question.

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
