# Peer review of "The Demographic Implication for Promoting Sponge City Initiatives in the Chinese Megacities: A Case of Wuhan"

_water, doi:10.3390/w14060883_

Round 1
Reviewer 1 Report
This paper dealt with solved problems using traditional and available methods. There is no novelty in this paper. I am concerned that the results of this study will not contribute to Sponge City Initiatives. My recommendation is therefore to reject this manuscript.
Author Response
Comment from Reviewer 1:
Reviewer 1
This paper dealt with solved problems using traditional and available methods. There is no novelty in this paper. I am concerned that the results of this study will not contribute to Sponge City Initiatives. My recommendation is therefore to reject this manuscript.
Response:
We regret to hear that the reviewer considers the paper lacks novelty. We will try explaining the ideas and the contribution briefly here.
We have used traditional methods in this investigation including the in-person survey, descriptive statistics and Chi-square tests to analyse the survey results. These are all traditional methods. We do not intend to claim novelty from the direction of developing a new methodology in investigating environmental perception.
Instead, we consider this manuscript contributes to the study of the Sponge City initiative from the interesting results based on these traditional analytical methods. In general, we identified that the demographic backgrounds affect how the citizen evaluate the environmental conditions, identify the environmental problem and judge the government initiative such as Sponge City Programme to manage the environmental issues. As not many participants were aware of the SPC programme, We also provided further analyses regarding how the demographic factors affected the use of media to access the information. Combining the two, specifically for Wuhan, a practical recommendation regarding the way to improve the dissemination of the information are provided. We also indicated that as the demography may change over time, with the development of social media, based on the results we observed, the means of policy dissemination may need to be updated as suit.
For further highlighting these points, we have updated the sections in the manuscript to better present the results of the effect of demographic factors on the use of media to access the Sponge City related information.
Additionally, during revising the manuscripts, we had another check of language and revised part of the expressions to hopefully clarify some confusion.
And we hope with the revision based on reviewers’ comments and suggestions, the manuscript will appear publishable.
Reviewer 2 Report
This is an interesting paper dealing, investigating public perception of Sponge City initiatives and related issues. Whilst the demographics of the survey respondents is skewed towards certain groups (younger, higher educated, etc), the relatively large number of respondents enables the impact of different factors on perception to be determined. The results are generally well presented and give some useful insights.
Minor issues:
- Introduction is fine but needs some references.
- Lit review covers a lot of area in not much detail; suggest expanding content or reducing scope
- Section 2.3 should be in section 3 as it is case study info
- Some recommendations to improve public understanding of the SCP are given in a number of places, but I think the paper would gain more traction if it included a more explicit “Recommendations” section in the conclusions section.
Reviewer 3 Report
Analyses are interesting. Some comments are listed below:
- Historical flooding events can be listed in a list for reference. After the floods, people tended to be concerned about the related news. In the list, the affected population and area can be listed. And what’s the coverage of the flooded affected area and the surveyed area. Some discussions may be made from those data.
- Some figures are recommended to draw in color for better reading. For example, “Minor Issue” and “Very Serious” of Fig. 2a are a little harder to read.
- One of the subgroups of “Age” is 18-19, any reason to define this group. The same issue of “Years of living in Wuhan” existed for 0-2 yrs, 2-5 yrs, etc.
- The population older than 50 years is 42.5%, but the sample size of this subgroup is only 9.3%. If possible, the age distribution of city central area is recommended to list for reference. Those limitations is recommended to describe in the “Discussion and Further Research.”
- To see if the “age” and the “Residential Time” correlate, to draw the scatter plot of the two is helpful.
- The “Media Public Received Information” 4.6 is curial disaster management. It’s recommended to say more on this topic. Some figures may be added. For example, the pie chart of “Media Public Received Information” and the breakdown for different ages. In lines 707-709, it said, “The decreased percentages among younger groups in using TV and newspapers were made up by the proportion who reported they obtained the information from the lectures (37%±7% for the group younger than 20; 31%±5% for the group between 20 and 30)”. Does it mean that many younger people get information from the class? It’s interesting to know if the younger people tend to get the information on the web.
- The consistence of words. For example, “Residential Time” or “Years of living in Wuhan.”
- Typically, “Conclusion” is separated from “Discussion and Further Research”. Therefore, I suggested removing the Discussion and Further Research to a new section.
Author Response
Please see the attachement
